# The Role of Uric Acid in Human Health: Insights from the *Uricase* Gene

**DOI:** 10.3390/jpm13091409

**Published:** 2023-09-20

**Authors:** Youssef M. Roman

**Affiliations:** Department of Pharmacotherapy & Outcomes Science, School of Pharmacy, Virginia Commonwealth University, Richmond, VA 23298, USA; romany2@vcu.edu

**Keywords:** neurodegenerative diseases, cardiovascular diseases, fructose metabolism, adaptation, evolutionary biology, genomics, gout, hyperuricemia

## Abstract

Uric acid is the final product of purine metabolism and is converted to allantoin in most mammals via the uricase enzyme. The accumulation of loss of function mutations in the *uricase* gene rendered hominoids (apes and humans) to have higher urate concentrations compared to other mammals. The loss of human uricase activity may have allowed humans to survive environmental stressors, evolution bottlenecks, and life-threatening pathogens. While high urate levels may contribute to developing gout and cardiometabolic disorders such as hypertension and insulin resistance, low urate levels may increase the risk for neurodegenerative diseases. The double-edged sword effect of uric acid has resurrected a growing interest in urate’s antioxidant role and the uricase enzyme’s role in modulating the risk of obesity. Characterizing both the effect of uric acid levels and the uricase enzyme in different animal models may provide new insights into the potential therapeutic benefits of uric acid and novel uricase-based therapy.

## 1. Background

Uric acid is the final product of purine metabolism in humans. Serum urate levels vary significantly between different species. Particularly, humans have the highest urate levels compared to all other mammals. Markedly high uric acid levels in humans have been caused by the loss of the uricase enzymatic activity during the Neogene period [1,2].

In humans, two-thirds of uric acid is excreted through the kidney and one-third through the gastrointestinal tract [3]. While humans do not produce uricase per se, a growing body of literature has suggested that the gut microbiome, a primary source of uricase, may have a role in compensating for the loss of the *uricase* gene [4,5,6,7]. Increased urate production or underexcretion of uric acid can increase serum urate levels beyond the solubility threshold, accelerating the formation of monosodium urate (MSU) crystals in and around the joints. Mobilization of MSU crystals could trigger an inflammatory response known as gout flares.

In mammals, uricase is the primary enzyme that converts uric acid to allantoin, a more water-soluble form. This comparative biology approach paved the road for developing the recombinant form of uricase to treat advanced tophaceous gout and prevent uricosuria in treating certain hematological and solid tumor malignancies [8,9,10,11]. Currently, rasburicase and pegloticase are the approved uricase recombinant proteins available for treating advanced gout or preventing urate nephropathy following chemotherapy. Rasburicase is a recombinant fungal uricase, while pegloticase is a mammal recombinant uricase. However, administering pegloticase is associated with a high risk of immunogenicity toward the pegylation moiety, causing the drug to be ineffective for some patients [12].

In most mammals, uric acid levels could range between 1 and 2 mg/dL [1]. Unlike mammals, primates have an increased urate level due to declining uricase activity, with healthy individuals maintaining uric acid levels around 3–4 mg/dL. While the loss of the *uricase* gene rendered humans the only mammal to develop gout, it was also proposed that retaining uric acid provided an adaptive mechanism for human survival [1,2,13,14]. Nonetheless, it has been elusive to characterize the exact adaptation mechanism, ushering in many hypotheses and speculations about the roles of thrifty genes and uric acid in human health.

Uricase is present in nearly all living organisms, from bacteria to mammals, with a common evolutionary origin. The gradual accumulation of genetic mutations, mainly nonsense mutations, in the *uricase* gene, during the evolution process, rendered the enzyme undetectable in humans and specific monkeys [15]. Specifically, the lack of uricase activity in humans has been greatly attributed to the nonsense mutation in codon 33 in exon 2, a nonsense mutation in codon 187 and a splice mutation in exon 3 [16]. The occurrence of independent mutations during hominoid evolution in parallel with the evolution of Old World and New World monkeys could be interpreted as evidence that there must have been a naturally selective advantage for early primates in having elevated uric acid [13,16].

The loss of the uricase enzyme in humans and the extensive reabsorption of uric acid suggest that retaining uric acid is essential to human health; hence, evolutionary physiology has treated uric acid not as a harmful waste. Also, this high conservation of uric acid has rendered humans susceptible to uric acid changes induced by diet and their propensity to develop gout spontaneously. Despite the well-established relationship between high uric acid levels and gout, researchers have been actively trying to elucidate the health benefits of uric acid, its role in adaptation to different environments, and potentially inform the development of less immunogenic uricase-based gout therapy [17].

Proposed theories about the role of uric acid have been previously described [1]. For example, growing evidence suggests that uric acid is a powerful antioxidant, accounting for roughly 50–60% of those naturally occurring in humans [18]. This naturally occurring protection may have elongated the life expectancy of hominoids (apes and humans). While two-thirds of uric acid levels are because of endogenous sources, one-third of uric acid is the result of external factors, mainly diet and certain lifestyle factors. Therefore, urate levels in hominoids are viewed as a mirror to the major and consequential changes in dietary patterns of early hominoids. 

It is hypothesized that the loss of the L-gulono-γ-lactone oxidase gene, which is involved in vitamin C biosynthesis, was driven by the significant ingestion of fruits and vegetables, a primary source of vitamin C. As a result, uric acid became the primary antioxidant and free radical scavenger when fruit and vegetable consumption dramatically decreased due to substantive climate changes [2]. Even though most vertebrates can make vitamin C, hominoids are one of the few species that cannot make it [19]. These observations suggest that retaining high uric acid levels may have enabled hominoids to garner survival benefits above and beyond the antioxidant activity of uric acid. A summary of the factors that may influence serum urate levels and their implications for human health are described in Figure 1.

Population dietary patterns and cultural habits have been implicated in the high prevalence of hyperuricemia and gout among specific population groups [20]. For example, the Māori of New Zealand have been long known to have genetic polymorphisms that could increase their risk of developing both hyperuricemia and gout. However, there is no mention of gout among the same population before the 18th century. Historically, the traditional diet of this population group included sweet potato, fern roots, birds, and fish. After the introduction of the Western diet in the early 1900s, an epidemic of both obesity and gout developed. To date, the Māori of New Zealand have one of the highest gout rates in the world [21,22].

The objective of this review article is to highlight the effects of high and low serum urate levels on human health. A review of the implication of genetics, as it intersects urate levels and various phenotypes associated with serum urates, is critical to fully characterize the contribution of different environmental changes to human health. Additionally, analyzing the ramifications of the loss of the uricase gene could expand our knowledge base to better characterize the various physiological roles that uric acid could play in human health and explain the current public health trends and population risk differences for common and chronic diseases.

## 2. Genetic Correlation between Urate Levels and Cardiometabolic Traits

Developing hyperuricemia or gout is multifactorial, including genetics, dietary habits, and other lifestyle factors [20]. Multiple studies have demonstrated that certain risk factors, such as high body mass index (BMI), male sex, advanced age, and increased consumption of meat, alcohol, and high-fructose corn syrup, could increase uric acid levels, increasing the risk of gout [3]. Other factors, such as genetic polymorphisms across major uric acid transporters, have been linked to uric acid levels and the risk of developing gout. For instance, genetic polymorphisms identified through genome-wide association studies (GWAS), including *SLC2A9*, *SLC16A9*, *SLC17A1*, *SLC22A11*, *SLC22A12*, *ABCG2*, *PDZK1*, *GCKR*, *LRRC16A1*, *HNF4A/G*, and *INHBC* are some of the major determinants in uric acid levels and modulators of gout risk (Table 1) [23,24]. Additionally, the genetic polymorphisms within the former genes do not only show disproportionate allele frequencies among different ethnic and racial groups, but their frequencies are directionally consistent with the epidemiology of hyperuricemia or gout within their respective population [25,26].

Serum urate levels are highly heritable, with an estimate of 30–60%. Despite the modest degree of heritability of urate levels, the variability explained in serum urate levels by constructing genetic models remains limited to up to 17% using SNPs at 183 loci [28]. In contrast, non-genetic models could explain a higher proportion of the variability in serum urate and precisely predict the risk of gout compared to traditional genetic risk models. In a study using UK Biobank data, the genetic risk model was a significantly weaker predictor (area under the receiver operating characteristic curve (AUROC) = 0.67) than the demographic model (age and sex) (AUROC = 0.84) [28]. Nevertheless, elucidating the presence of genetic correlation may explain the shared biology of traits or diseases. Furthermore, leveraging quantitative genetic approaches can deconstruct the bivariate association patterns observed at the phenotypic levels into genetics and environmental components. The same framework has been used to disentangle the phenotypic correlation between the genetic and environmental causes of serum urate and traits representing comorbidities of clinical importance, such as serum creatinine, blood pressure, serum blood glucose, and BMI [30]. Data from two independent datasets indicated independent genetic overlap between urate level and creatinine and urate and metabolic syndrome, supporting the notion that genetics may partly explain the clustering of gout, chronic kidney disease, and other metabolic comorbidities [30].

Multiple major epidemiological studies repeatedly showed a strong association between serum urate levels with multiple cardiometabolic traits and other cardiovascular risk factors [31,32,33]. Using the National Health and Nutrition Examination Survey, the conditional probability of obesity, given the individual has hyperuricemia, is between 0.4 and 0.5 [34]. Similarly, the prevalence of hypertension, diabetes, and chronic kidney disease proportionally increases as uric acid levels increase; the same risk factors are more common in patients with gout than those without gout [35]. With a robust genetic underpinning, the genetic correlation between urate levels and complex diseases has been evaluated [28,30]. Tin et al. assessed the genetic correlations between urate and 748 complex traits using cross-trait linkage disequilibrium score regression. The serum urate levels were significantly genetically correlated with 214 physical traits and health conditions. The highest positive genetic correlation was with gout, followed by the trait of the metabolic syndrome [28]. The largest negative genetic correlations observed included high-density lipoprotein and estimated glomerular filtration rate consistent with the observational associations from epidemiological studies. The same group also assessed whether genetic correlations reflect a causal or pleiotropic relationship. To this extent, the genetic causality proportion (GCP) of seven common cardiometabolic traits was examined, using the genetic causality of urate on gout as a positive control. Among the seven examined phenotypes, the largest GCP was observed for adiposity-related phenotypes (e.g., waist circumference) [28]. Nevertheless, the directionality of the effect (high urate causing obesity vs. obesity causing high urate) is the focus of Mendelian randomization studies, suggesting a causal effect of obesity on urate levels [36]. This directionality of effect is consistent with the hypothesis that increased waist circumference, a primary obesity indicator, represents a high reservoir of purine, the precursor for uric acid [28].

## 3. Uricase Activity: Fructose Metabolism and Energy Storage

The slow decline in uricase activity has been linked to climate and vegetation conditions associated with food scarcity and famine periods. It is hypothesized that the progressive decrease in uricase activity enabled our ancestors to readily accumulate and store fat via the metabolism of fructose to fructose-1-phosphate from fruits in anticipation of significant climate changes or droughts [15]. Additionally, increased uric acid levels because of declining uricase activity could have amplified the effect of fructose on energy and fat storage (Figure 2).

While high serum urate levels may enhance the development of metabolic syndrome, uric acid could further potentiate the metabolism of fructose by increasing the expression of fructokinase, which in turn can increase the formation of fructose-1-phosphate and the formation of energy storages such as glycogen and triglycerides (Figure 2) [37]. A critical survival aspect is having sufficient food, water, and essential electrolytes to maintain body mass and normal physiological functions. While having external storage of these nutrients could serve as a short-term solution, this approach is unsustainable as a long-term survival mechanism, given the unpredictable nature of climate change and the potential risk for these storages to corrupt. Alternatively, intrinsic changes (i.e., survival switch) in bodily physiological functions present an adaptive survival mechanism to store nutrition and energy resources as a reservoir when food scarcity occurs (Figure 3).

Evidence from animal research has demonstrated excessive consumption of fructose sources in preparation for food, oxygen, and water scarcity. Fructose ingestion has been shown to conserve water by stimulating vasopressin, which reduces water loss through the kidney and stimulates fat and glycogen production as a source of metabolic water [38]. Additionally, fructose consumption may stimulate thirst, which can act as a mechanism to increase the water content [14,38,39]. Vasopressin also contributes to fat accumulation, which could provide metabolic water when metabolized [39].

Moreover, fructose could induce insulin resistance, partly due to glucogenesis mediated by fructokinase-dependent activation of the AMP deaminase pathway with the generation of uric acid (Figure 2) [40]. With glucose being the primary source of immediate energy needs, especially the brain, preserving an adequate supply of glucose is crucial for survival. Fructose reduces resting metabolism and stimulates fat and glycogen accumulations to reduce metabolism and maintain glucose levels, which could explain the insulin resistance induced by fructose (Figure 3) [14]. These metabolic processes could have enabled primates and humans to survive starvation during significant climate changes, including restricted water access, droughts, or devegetation.

Natural selection for losing uricase activity has enabled hominoids to retain high uric acid levels, which may have enhanced fructose metabolism and increased fat stores. Combined, increased uric acid levels because of the lost uricase activity might have provided a survival advantage during those critical historical times. These harsh environments may have ushered in the introduction of biological mutations that improved survival in the setting of starvation. The same notion may be responsible for the current obesity epidemic, where food is plentiful and global fructose consumption is rising [40]. These observations may collectively support the thrifty genes hypothesis: individuals who could easily store extra energy would have had an evolutionary advantage during famines and partly explain the global rise in obesity and metabolic syndrome [41,42,43].

## 4. Regulating Blood Pressure: The Role of Salt and Uric Acid

A direct relationship between salt intake and blood pressure (BP) has been well established. Excessive sodium intake has been shown to increase blood pressure and the onset of hypertension. Increased sodium consumption is associated with increased water retention, peripheral vascular resistance, oxidative stress, and microvascular remodeling [44]. Following this growing evidence, salt restriction has become a pillar in hypertension management and patient counseling. However, the BP response to changing salt intake displays marked inter-individual variability. Therefore, salt sensitivity could be viewed as a continuous parameter at the population level. Inter-individual variability in the BP response to salt modifications could be compounded by the other biochemicals that exert varying effects on blood pressure, such as potassium and uric acid levels.

While being a naturally occurring antioxidant, uric acid has been strongly implicated in developing the hypertension epidemic and maintaining blood pressure during the early hominoids. Fossil evidence suggests that early hominoids heavily relied on fruits for their dietary needs. With a fruit-based diet, the sodium intake during the Miocene period was expected to be very low to maintain blood pressure. To evaluate this hypothesis, mild hyperuricemia was induced in rats by inhibiting uricase, recreating the pre-historic conditions by feeding the rats a low-sodium diet. Compared with hyperuricemic rats, normal rats receiving a low-sodium diet experienced either no change or a drop in systolic blood pressure. In contrast, systolic blood pressure increased in hyperuricemic rats, proportional to uric acid levels, and could be mitigated using allopurinol [45,46]. Therefore, it was presumed that increased uric acid levels may have maintained blood pressure during low sodium intake.

Uric acid levels may also contribute to the salt sensitivity associated with developing hypertension. A study using rats treated with oxonic acid and a low-salt diet for seven weeks resulted in mild hyperuricemia compared with controls. Following the discontinuation of oxonic acid, uric acid levels did not differ between groups. However, the renal tissue obtained from rats with hyperuricemia showed persistent arteriolopathy with decreased lumen diameter and mild interstitial inflammation. When rats were then randomized to a low- or high-salt diet, an increase in blood pressure from the high-salt diet was observed only in rats that had been previously hyperuricemic [13]. Collectively, a potential hypothesis would be that salt-sensitivity-induced hypertension could be mediated by uric acid levels.

## 5. Hyperuricemia and Hypertension: A Cause or Effect?

Inflammation remains the principal suspect linking gout and cardiovascular diseases. Mechanisms whereby serum urate levels could contribute to the development of hypertension are partly related to the uric acid’s primary effect on the kidney. These mechanisms include activating the renin–angiotensin–aldosterone system (RAAS) and the disposition of urate crystals in the urinary lumen. Additional evidence suggests that elevated serum uric acid can decrease nitric oxide production, causing endothelial injury and dysfunction. It is well established that monosodium urate crystals can activate the NLRP3 inflammasome cascade and the production of inflammatory cytokines such as IL-1 beta and IL-8 [47]. Although uric acid is predominantly eliminated through the kidney, new evidence has reported that urate crystals can deposit in the coronary artery among gout patients [48]. The deposition of urate crystals in the aorta and main arteries was associated with a higher coronary calcium score and can trigger an inflammatory response like the observed response in the kidney. Beyond the joints, these observations support the hypothesis that high serum urate levels could deposit into soft tissues and extraarticular regions, which can then become pro-oxidants, increasing the inflammatory burden and the development of the multiple comorbidities observed in patients with gout or hyperuricemia [49].

A reverse causality of hypertension and hyperuricemia is also possible. Patients treated for hypertension could develop hyperuricemia secondary to decreased renal blood flow, increased BMI, and concomitant use of medication that reduces uric acid excretion or increases uric acid production. Another proposed hypothesis suggests that, given the route of elimination, uric acid can directly cause renal injury, which could result in the activation of RAAS, causing hypertension [47].

Hypertension is an epidemic affecting more than 30% of adults worldwide [50]. While genetics may contribute to the development of hypertension, the contribution has been limited, suggesting that the external environment plays a significant role in developing hypertension compared with genetics. Moreover, a growing body of evidence has repeatedly reported on the association between elevated urate levels (hyperuricemia) and hypertension. For example, a cross-sectional study determined that each 1 mg/dL increase in serum urate contributes to a 20% increased prevalence of hypertension in a general population not treated with hyperuricemia and hypertension [51]. A retrospective study also identified that having hyperuricemia was independently associated with the development of hypertension, with 95% confidence interval hazard ratios of 1.37 (1.19–1.58) in men and 1.54 (1.14–2.06) in women [52].

Hyperuricemia is an independent risk factor for multiple cardiovascular disease risk factors, which can, collectively, elevate the risk of cerebrovascular events. Nonetheless, the direct role of uric acid levels in developing stroke events is inconclusive [53,54]. On the contrary, increased uric acid levels during acute ischemic stroke events may result in better functional treatment outcomes than controls, supporting the antioxidant effects of uric acid [55,56].

The incidence and prevalence of hypertension and obesity were minimal in specific populations until Westernization and the adoption of a Western diet [57,58]. Additionally, the immigration of select population subgroups, Filipinos and Japanese, has been linked to earlier gout onset, higher gout and hypertension rates, and higher mean urate levels than their native home residents [25,58,59,60,61,62]. These observations support the hypothesis that dietary changes from a traditional diet to a Westernized diet enriched with fatty meats may be responsible for the increased prevalence of gout, high serum urate levels, and increased incidence of hypertension and diabetes.

The underlying association mechanism between westernization and the development of hypertension is complex. However, the most likely cause is fueled by the acculturation and assimilation of the Western world, which brings multiple changes in dietary habits and lifestyle factors [57]. For instance, foods with high sodium and low potassium content will likely increase blood pressure. A sedentary lifestyle is also associated with low energy expenditure, which could aggravate blood pressure secondary to dietary changes.

## 6. Uric Acid and Neurodegenerative Diseases: The Antioxidant Hypothesis

The old hypothesis that the chemical structure of uric acid could mimic specific brain stimulants, such as caffeine or methylxanthine, may have provided selective advantages to early hominoids [63]. To this hypothesis, the loss of uricase activity in early hominoids has caused an increase in uric acid levels in the brain, which could have given rise to quantitative and qualitative intelligence among early hominoids. Indeed, limited studies have shown that urate plasma levels may be associated with higher intelligence [63,64,65]. Nonetheless, the level of intelligence is a complex multifactorial trait involving both environmental and physiological factors.

Multiple epidemiological studies have examined the association between serum urate levels, gout diagnosis, and brain-related diseases [66,67]. Specifically, the associations between urate levels or gout and Alzheimer’s disease (AD) and Parkinson’s disease (PD) have been a focal point in recent studies [68,69,70]. However, the results of these studies have been conflicting or showing no association [71,72]. Partly, these inconclusive findings have been attributed to the study design, not accounting for uric acid level confounders, and the varying degrees of disease progression among study subjects [73]. While the transport of uric acid into the brain remains elusive, uric acid is produced in the human brain [69]. Urate production in the brain has paved a path for a great deal of research into whether uric acid could exert antioxidant activity, protecting the brain from developing neurodegenerative diseases by mitigating the burden of oxidative stress sequelae. Uric acid can function as an antioxidant and block peroxynitrite, which could be beneficial in slowing the progression of PD and multiple sclerosis (MS) [74,75,76]. Lower uric acid levels have been linked with a greater risk of developing PD, the severity of motor features, and faster progression of both motor and non-motor features [69]. This observation has led to the hypothesis that raising uric acid may provide a neuroprotective effect in PD patients [77]. However, the study failed to show any benefits from raising uric acid, suggesting that PD may lower uric acid or that uric acid is a biomarker of other processes relevant to PD [72].

To assess the causality between uric acid and PD, a Mendelian randomization approach was used to systematically evaluate the inherent risk of lower urate levels and disability requiring dopaminergic treatment in early PD patients. Genotyping 808 patients for three loci across the *SLC2A9* (rs6855911, rs7442295, and rs16890979), cumulative scores of the risk alleles were created based on the total number of minor alleles across the three loci [78]. Serum urate levels were 0.69 mg/dL lower among individuals with ≥4 *SLC2A9* minor alleles versus those with ≤2 minor alleles [78]. A 5% increased risk of PD progression with a 0.5 mg/dL decrease in serum urate was observed. However, the hazard ratio for the progression to disability requiring dopaminergic treatment, though increased with the *SLC2A9* score, was not statistically significant (*p* = 0.056) [78].

Another study evaluated the association of a single nucleotide polymorphism (rs1014290) within *SLC2A9* and PD in the Han Chinese population. Using a case–control study design, the serum urate levels were significantly lower in PD patients than in controls. Individuals with the rs1014290 TT and CT genotypes had significantly higher serum urate levels than the CC genotypes [79]. The lower serum urate-associated allele/genotype (C/CC) was statistically more frequent in PD patients than in controls. Those with the CC genotype had significantly higher odds of PD than those with TT or TC. Collectively, these data raise the possibility that modulating the *SLC2A9*-encoded transporter (GLUT-9) might be an alternative approach for urate-elevating therapy, compared to urate precursor administration as a potential PD treatment target.

The role of uric acid levels in mild cognitive impairment (MCI), a common diagnosis that precedes the development of clinical AD, has been extensively investigated. While asymptomatic hyperuricemia can go undiagnosed and rarely treated, exploring the association between hyperuricemia and neurodegenerative diseases may be confounded by recall bias, disease misdiagnosis, and secondary causes of hyperuricemia. Nonetheless, patients with gout are considered to have the highest uric acid burden and represent the extreme phenotype of hyperuricemia. Therefore, the association between gout diagnosis and the risk of neurodegenerative diseases has been of significant interest in neurodegenerative disease research.

Interest has emerged in investigating the association between serum urate levels and mild MCI. Given the transitional stage between declining cognition and a clinical diagnosis of AD, clinical interventions targeting MCI can become a window of opportunity to arrest or slow the progression of the disease [80]. To this end, the neuroprotective effect of uric acid on MCI has been evaluated in multiple cross-sectional studies [81,82,83,84,85]. While the results of some of these studies appear conflicting, they indeed highlight the double-edged sword nature of uric acid and the potential of other confounders that may affect urate levels [86].

Multiple studies have shown that subjects with AD have lower serum urate levels than matched controls. These findings have ushered in the hypothesis of uric acid being a neuroprotective compound and potentially a therapeutic target in neurodegenerative diseases. However, lower brain urate levels in AD patients than matched controls could be owing to the reduced metabolic activities that might lead to reduced uric acid production. To this extent, urate levels may serve as a proxy for overall brain metabolic activity rather than being the culprit of the disease. Additionally, one of the significant serum urate predictors is the overall nutritional status and BMI. Clinical manifestations of AD are often preceded by substantive weight loss, which may significantly account for the lower serum urate levels in the AD presentation.

Generally, patients with hyperuricemia tend to have a higher cardiovascular disease burden, warranting a concomitant use of varying medications that may affect serum urate levels. Also, serum urate levels can be influenced by sex, obesity, genetics, and overall nutrition. Therefore, the measurement of uric acid may not accurately reflect the inherent urate level of the individual to draw a robust association between serum urate levels and the condition of interest, let alone the cross-sectional design of these studies. While uric acid may serve as a potent antioxidant, a robust epidemiological study designed to identify the optimal urate levels to maximize the benefits and minimize the risks is needed.

## 7. Hyperuricemia and Innate Immune System: Acquired Protection?

Selective advantages to developing hyperuricemia in distinct population groups may have been driven by natural selection. This natural selection advantage may partly explain the inherently increased predisposition for hyperuricemia or gout among select population groups, especially when challenged by unprecedented environmental threats or culturally altering events. Besides being a naturally occurring antioxidant, urate levels may have enabled humans to survive crucial environmental stressors and pathogenic threats. When crystallized, urate may trigger the innate immune response. For this reason, monosodium urate crystals are considered a critical natural endogenous immune response trigger. With infectious diseases being one of the strongest drivers of natural selection, it is presumed that genes associated with raising urate may have been selected during major depopulation events. This theory may partly explain the genetic predisposition for hyperuricemia/gout and the higher mean uric acid levels in the Pacific and Oceania regions compared with whites and the Western world [25,87,88,89,90].

Historically, exposure to pathogenic or parasitic infections such as malaria has pressured the human genome of distinct populations to develop disorders like sickle-cell disease and glucose-6-phosphate dehydrogenase deficiency as a survival switch. Therefore, it is plausible that the inherent risk of hyperuricemia could be considered another adaptation to protect against malaria or other pathogens that threatened our ancestors. Urate levels released from erythrocytosis because of malaria infection have been shown to activate the host’s inflammatory response, facilitating the detection of the parasite [87]. Given the role of enhancing the innate immune response [91], it is permissible that genotypes promoting increased urate levels were positively selected for and during stressful environments or significant depopulation because of infection or threatening pathogens [25,27,62,92].

## 8. Uric Acid: A Therapeutic Target or Disease Bystander?

Uric acid levels are the strongest predictor of developing gout via the deposition of monosodium urate crystals in and around the joints. The chronic suppression of uric acid levels is the hallmark of reducing the MSU crystal burden and preventing future gout flares. While different treatment modalities have been developed to lower serum urate, most treatments are focused on inhibiting uric acid production. In the advanced forms of gout, the recombinant form of uricase has been used to reduce the tophi burden or if the patient is not responsive to available treatments. In addition to gout, high uric acid levels have been incriminated in different chronic disease states, including hypertension, metabolic syndrome, diabetes, non-alcoholic fatty liver disease, and chronic kidney disease [32,41,93,94].

While numerous experimental and clinical studies support the role of uric acid as an independent cardiovascular risk factor, a handful of studies also suggested that lowering serum urate may improve blood pressure. These studies included pre-hypertensive obese [95], hypertensive adolescents [96], hypertensive children on an angiotensin-converting enzyme inhibitor [97], and adults with asymptomatic hyperuricemia [98,99].

A growing body of evidence also suggests that urate-lowering therapy does not affect blood pressure. A crossover single-center study enrolled ninety-nine participants randomized to 300 mg of allopurinol or placebo over four weeks [100]. There was no difference in the change in blood pressure between the groups; however, the allopurinol group had a decrease in uric acid and improved endothelial function, estimated as flow-mediated dilation [100]. What is noteworthy is that the study participants were relatively young (28 years old), had relatively normal systolic blood pressure (123.6 mm Hg), and had a relatively normal uric acid level at baseline (5.9 mg/dL) [100].

In another double-blind placebo-controlled trial, overweight or obese participants (*n* = 149) with serum uric acid ≥5.0 mg/dL were randomly assigned to probenecid, allopurinol, or placebo [101]. The primary endpoints were kidney-specific and systemic renin–angiotensin system (RAS) activity. Secondary endpoints included mean 24 h systolic blood pressure, mean awake and asleep blood pressure, and nocturnal dipping. The trial found that lowering uric acid did not affect kidney-specific or systemic RAS activity after eight weeks or mean systolic blood pressure [101].

With gout and hyperuricemia being associated with increased sterile inflammation, earlier reports suggested that the use of allopurinol in patients with hyperuricemia can significantly reduce inflammatory biomarkers [98]. A recent study showed that urate-lowering therapy, mainly allopurinol, was significantly associated with lower hs-CRP, LDL, and total cholesterol levels in patients with gout compared with those not receiving urate-lowering therapy [102]. These results support that patients optimally treated for gout can garner added benefits above and beyond reduced uric acid burden. Although urate-lowering therapy was not associated with lowering blood pressure, a high chronic kidney disease diagnosis in patients receiving allopurinol was observed [102]. Overall, the study suggested that uric acid may be a remediable risk factor to lower the atherogenic disease burden in patients with gout [102].

Inflammatory reactions are thought to be crucial contributors to neuronal damage in neurodegenerative disorders such as AD, PD, MS, and amyotrophic lateral sclerosis (ALS) [75,76]. Among the toxic agents released in brain tissues by activated cells is peroxynitrite, the product of the reaction between nitric oxide and superoxide [76]. As a potential peroxynitrite blocker, raising urate levels has been recently explored in neurodegenerative disease management [74]. Leveraging the neuroprotective hypothesis of uric acid, administering a uric acid precursor, inosine, was evaluated in prospective clinical trials.

In patients with early PD, administering inosine was safe, tolerable, and effective in raising serum and cerebrospinal fluid urate. However, among patients recently diagnosed with PD, treatment with inosine compared with placebo did not result in a significant difference in the rate of clinical disease progression, questioning the use of inosine as a treatment for early PD [77,103]. Another study evaluated the safety and tolerability of elevated uric acid levels in patients with amyotrophic lateral sclerosis (ALS). In a 12-week study, inosine use was shown to significantly increase serum urate levels in patients with ALS safely and effectively [104]. Similarly, a 20-week study showed that inosine significantly increased uric acid without treatment-emergent adverse events compared to placebo. However, the study did not show functional benefits associated with inosine in ALS patients compared with receiving a placebo [105].

Despite the disappointing results of the urate-elevating studies, it is critical to recognize that of the above studies, some lacked the sample size to detect adequate differences. Additionally, inosine use could be problematic in achieving the desired serum urate levels as it can enter the purine salvage pathway. While addressing the sample size issue for future studies is obvious, the mechanism to elevate uric acid requires a robust thought process and potentially targeting uric acid excretion transporters.

## 9. Current and Future Perspectives

The loss of activity mutations in the uricase gene may have been protective in situations of famine and food scarcity; thus, it rapidly took over the ancestral population, likely driven by harsh environmental factors during the Miocene era. In modern times, this adaptation is possibly leading to a state of hyperuricemia, increasing the risk of gout and other cardiometabolic disorders. Today, all humans are considered uricase knockouts. Loss of the uricase enzyme resulted in the inability to regulate uric acid effectively. The adaptation to famine and food scarcity in early hominoids dictated crucial changes to obtain and conserve energy for survival. In contrast to animal experiments and observational studies, targeting uric acid levels for therapeutic purposes beyond the purpose of gout remains conflicting.

As hyperuricemia and gout animal models continue to be instrumental in advancing the field of gout and uric acid metabolism, it is equally important to recognize that animal models are inherently predisposed to eliminating uric acid. Therefore, the effects of raising uric acid levels in animal models may not reflect the outcomes seen in human studies. Additionally, the sex effect on uric acid levels reinforces the need for adequate and robust analyses by sex. Studies have shown that women are more sensitive to the cardiometabolic effects of increased uric acid after menopause versus men. Uric acid levels may also present as a J-shaped phenomenon. Therefore, careful study design and a robust rationale for classifying or categorizing uric acid levels are needed to minimize the effect of extreme uric acid levels.

While genetics play a significant role in developing hyperuricemia or gout, it is equally important to recognize that the same genetics are therapeutic targets to modify uric acid metabolism. Genetic polymorphisms have been implicated in the racial difference in gout prevalence and may also contribute to the heterogeneity in response to urate-lowering or urate-elevating therapies. Therefore, genetic investigations into uric acid metabolism in clinical trials need to be considered in designing the study. Finally, as described earlier, uric acid levels are the culmination of endogenous cellular processes and external dietary sources; therefore, accounting for dietary and lifestyle habits in uric acid levels may minimize the effect of study confounders.

## Figures and Tables

**Figure 1 jpm-13-01409-f001:**
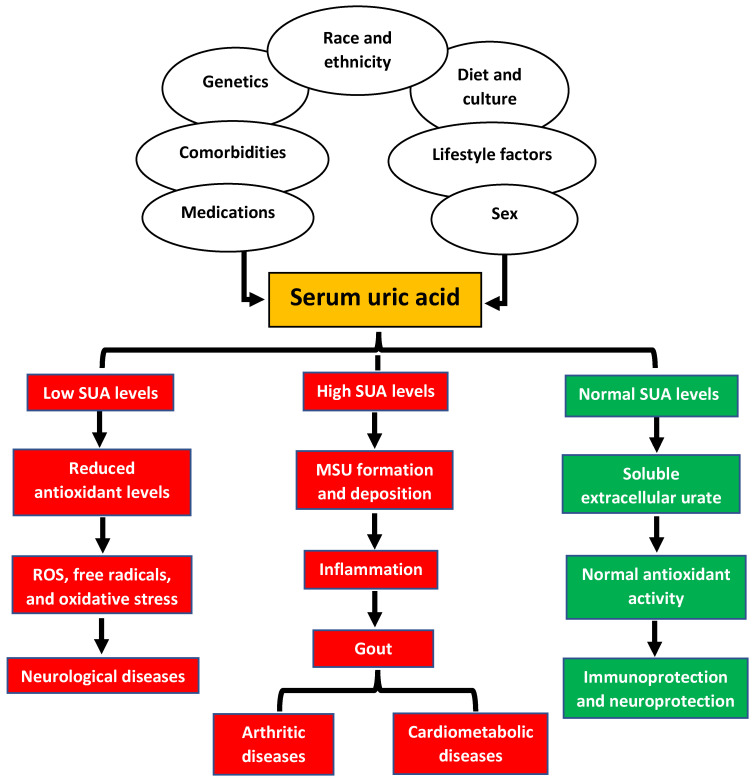
The determinants of serum uric acid levels and their implication for human diseases. SUA: serum uric acid; ROS: reactive oxygen species; MSU: Monosodium urate.

**Figure 2 jpm-13-01409-f002:**
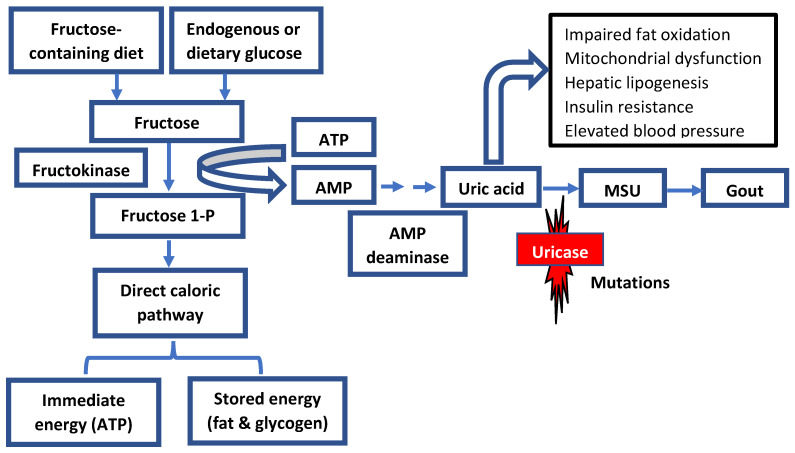
Role of fructose metabolism in fat and energy storage in humans. The loss of uricase activity could amplify the role of uric acid in response to fructose. Both fructose and uric acid would have led to the activation of the biological response to starvation, leading to increased fat and energy storages. Fructose-1-P: Fructose-1- Phosphate; AMP: Adenosine monophosphate; MSU: Monosodium urate.

**Figure 3 jpm-13-01409-f003:**
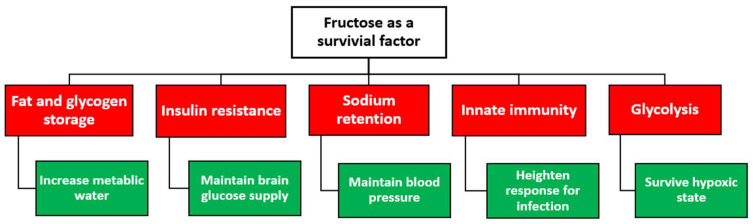
The proposed role of fructose as a survival factor in response to starvation in early hominoids.

**Table 1 jpm-13-01409-t001:** Summary of major urate regulation genes.

Gene	Protein	Possible Functions	References
*ABCG2*	ATP binding cassette subfamily G member 2 (ABCG2)	Regulating renal and gut excretion of urate. Gene polymorphisms have been strongly linked to urate underexcretion and the risk of early-onset gout in men. Genetic polymorphisms may also influence the therapeutic response to allopurinol and rosuvastatin.	[23,26,27,28]
*GCKR*	Glucokinase regulator (GCKR)	Regulatory protein that inhibits glucokinase in the liver and pancreatic islet cells by forming an inactive complex with the enzyme. Gene polymorphisms have been associated with fasting glucose, maturity-onset type-2 diabetes, hyperuricemia, and gout.	[23,26,28]
*LRRC16A*	Capping protein regulator and myosin 1 linker 1 (CARMIL1)	Cytoskeleton-associated protein. Gene polymorphisms have been associated with urate concentrations and gout subtype.	[23,26]
*PDZK1*	PDZK domain-containing scaffolding protein	Mediates the localization of cell surface proteins and plays a critical role in cholesterol metabolism. Gene polymorphisms have been linked to dyslipidemia, hyperuricemia, and gout.	[23,24,26,28]
*SLC2A9*	Solute carrier family 2 member 9 (GLUT9)	Regulating renal uric acid reabsorption. Gene polymorphisms have been linked to the risk of gout, especially in women.	[23,24,26,28]
*SLC16A9*	Solute carrier family 16 member 9 (MCT9)	Regulating monocarboxylic acid transporter. Gene polymorphisms have been linked to uric acid concentrations.	[23,26]
*SLC17A1*	Solute carrier family 17 member 1 (NPT1)	Sodium phosphate cotransporter. Gene polymorphisms have been linked with hyperuricemia and gout.	[23,26]
*SLC22A11*	Solute carrier family 22 member 11 (OAT4)	Urate reabsorption transporter. A target for some uricosuric drugs. Gene polymorphisms have been associated with hyperuricemia.	[23,26]
*SLC22A12*	Solute carrier family 22 member 12 (URAT1)	Uric acid reabsorption transporter. A major target for uricosuric drugs. Gene polymorphisms have been associated with hyperuricemia and gout. Loss of function in the gene can also lead to hypouricemia.	[23,26]
*NRXN2*	Neurexin 2	Member of the neurexin gene family that serves as a cell adhesion molecule. Genetic polymorphisms were associated with urate concentrations in Europeans and Chinese.	[24,26]
*INHBC*	INHBC	Member of the transforming growth factor ß superfamily that may inhibit activin A signaling, affecting a variety of biological functions including pituitary hormone secretion and insulin secretion.	[24,26]
*HNF1A*	Hepatic nuclear factor 1A	Encodes a transcription factor and enhances the promoter activity of PDZK1, URAT1, NPT4, and OAT4 in the human proximal tubule.	[28]
*HNF4A*	Hepatic nuclear factor 4A	Encodes another nuclear receptor and transcription factor that controls the expression of multiple other genes. It was shown to regulate the expression of *SLC2A9* and other members of the urate transportome.	[28]
*HNF4G*	Hepatic nuclear factor 4G	Encodes other transcription factors. Genetic polymorphisms are significantly associated with increased urate concentration in Europeans and gout in Chinese.	[24,29]

## Data Availability

Not applicable.

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
