# Peer review of "The Role of Uric Acid in Human Health: Insights from the Uricase Gene"

_jpm, 2023, doi:10.3390/jpm13091409_

Round 1

Reviewer 1 Report

The review article by Youssef M. Roman is focused on the effects of serum urate levels on human health, its relations to various diseases and the potential therapeutic benefits of modifying uric acid levels. The topic of this article is of interest to a broad audience including physicians, physiologists, pharmacologists, and evolutionary biologists. Primate during evolution progressively lost the uricase enzyme activity and the lack of this enzyme activity has been proposed to be advantageous as uric acid in serum plays various roles, including antioxidation, maintaining water in the body, inducing immune response, which could be beneficial to surviving from pathogen infection, and stimulating fructose metabolism and fat accumulation, which is believed to be important historically for hominoids to survive during climate changes and famines. By thoroughly reviewing literature, the author stated that abnormal levels, both high and low levels of uric acid may cause different human diseases thus modifying the level of uric acid, the end-product of purine metabolism is indeed a double-edged sword. High uric acid level has been well known correlating with gout disease and a growing population of hyperuricemia and gout has been a concern of public health. This is also linked to disorder of sugar and lipid metabolism resulting in cardiovascular diseases and obesity. Low level of uric acid is not as well-known for its negative effect on human health. However, it has been linked to high risk of neurodegenerative disease such as Parkinson’s disease and cognitive impairment in Alzheimer’s disease, although the mechanism has not been well elucidated.

As the author stated “serum urate levels can be influenced by sex, obesity, genetics, and overall nutrition. Therefore, the measurement of uric acid may not accurately reflect the inherent urate level of the individual to draw a robust association between serum urate levels and the condition of interest...”, the manuscript appears suited for the theme of “personal medicine”. However, the manuscript could be significantly improved prior to recommended publication in Journal of Personalized Medicine.

Major issues:

1.      The review article should include figures or tables for clearer presentation.

For example, a figure/table to illustrate various hypotheses, benefits and underlying mechanisms regarding the progressive loss of uricase during evolution of primates would be a nice addition.

Results of the GWAS study that revealed major determinants of uric acid levels and modulators of gout risk could be summarized in a table listing the gene names and encoded enzymes/ proteins and underlying mechanisms. As the manuscript in its current form, the gene acronyms listed without definition/explanation are not informative. “SLC2A9, SLC16A9, SLC17A1, SLC22A11, SLC22A12, ABCG2, PDZK1, GCKR, LRRC16A1, HNFA4G, and INHBC”

The complex relationship between fructose metabolism and loss of uricase/uric acid levels and underlying mechanisms could be presented as a figure.

What human diseases/disorders and how they are linked to either high or low levels of uric acid can be better summarized in a table.

2.      The paragraphs regarding fructose metabolism is very confusing as it is currently written, with several observations/hypotheses mixed and messed up together.

How uric acid stimulates fructose metabolism and hence adipose tissue formation is not clearly stated (Lanaspa et.al PLoS One. 2012;7(10):e47948.). Indeed, it acts through the transcriptional upregulation of the fructokinase. And this should be used as part of the evidence for hominoids employing loss of uricase as a survival mechanism for fat accumulation during famine period.

The author stated “Fructose induces insulin resistance partly due to glucogenesis mediated by fructokinase-dependent activation of the AMP deaminase pathway with the generation of uric acid”. This does explain why excessive fructose intake is linked to high uric acid level. However, this statement is placed in a wrong paragraph where the author wants to interpret the reason behind loss of uricase. Please note that we have already lost uricase enzyme activity millions of years. Nevertheless, excessive fructose intake has raised broad concern and it is generally believed to be a cause of hyperureicemia and gout. thus it is worth detailing the biochemical mechanism how even without uricase this would raise the uric acid level by increasing the rate of purine degradation through the activities of fructokinase and AMP deaminase.  

“Indeed, it is believed that with loss of the vitamin C encoding gene due to significant ingestion of fruits and vegetables, the primary source of vitamin C, uric acid became the primary antioxidant and free radical scavenger when consumption of fruits and vegetables dramatically decreased due to substantive climate changes.”-----This sentence is not comprehensible once again this is a mixture of different evolutionary events and hypotheses. Please rephrase it. Despite most vertebrates capable of making vitamin C, hominoids are not the only ones that can not make it. i.e, teleost fishes, guinea pigs, and some bats. Do all these vertebrates use uric acid as a radical scavenger? Did they also encounter climate changes and food shortage and lose their uricase?

3.      Diet has been attributed several times to gout onset throughout the text, more important than genetics in Line 70 for Maori of New Zealand who has gout susceptible alleles, more important than local environments in line 246 for Phillipinos and Japanese (compared with native residents). Diet including fructose, fatty meats and salt have all been mentioned. All gout-related diet information scattered throughout the text and appeared very disorganized. Please present it in a more thoughtful way.  

4.      As the author wrote in the abstract “…provide new insights into the potential therapeutic benefits of uric acid and novel uricase-based therapy.” Surprisingly, there is no discussion of krystexxa, or any other uricase-based therapy at all in the text.

For the purpose of increasing uric acid, the author spent quite a large volume on SLC2A9 without any definition/description of the gene and the coded transporter until at the very end defining it as GLUT9. I would recommend a clearer definition and description of the transporter and its functional relationship with URAT1 and how both are targets of the anti-gout drug, benzbromarone. So to increase uric acid level, one would likely need an activator of GLUT-9? is such a compound already available?  The authors listed many determinants of uric acid levels. SLC2A9, SLC16A9, SLC17A1, SLC22A11, SLC22A12, ABCG2, PDZK1, GCKR, LRRC16A1, HNFA4G, and INHBC…. Could they serve as targets to increase or decrease uric acid levels? 

Other than urine through kidney, gut provides the excretion pathway for the remaining 1/3 of uric acid. A recent paper described a gene cluster in human gut microbiome that compensates for loss of uricase (Liu et al., 2023, Cell 186, 3400-3413). Another research identified the long-sought anaerobic purine degradation pathway that bypasses the formation of uric acid (Tong et al., 2023, Cell Chemical Biology 30, 1-11). Given the importance of diet-derived purine, probiotics engineered with this pathway is potentially an efficient modifier of the uric acid level of the serum. Both papers should be cited.

Minor changes requested:

1.    “Markedly high uric acid levels in humans have been caused by the loss of the uricase-encoding gene during the Neogene period” Specifically, the lack of uricase activity in humans has been greatly attributed to the nonsense mutation in codon 33 in exon 2, a nonsense mutation in codon 187, and a splice mutation in exon 3.”

These two sentences contradict with each other.. “loss of the uricase-encoding gene” this expression is not accurate… I would suggest loss of the enzyme/enzyme activity.

2.  The definition of an abbreviation should be made in its first appearance and then the abbreviation can be used in later paragraphs. for example after definition, “monosodium urate” does not need to be spelled out and can be replaced by MSU

3.  vitamin C encoding gene” ---this is a wrong expression.. genes encode proteins i.e L-gulono-γ-lactone oxidase that is involved in vitamin C biosynthesis.

4. “cerebrovascular events” including “acute ischemic stroke events” may lead to the development of a neurodegenerative-like condition. But stroke is not classified as neurodegenerative disease.

A few sentences need to be rephrased, whihc have been pointed out in the comments above.

Author Response

The review article by Youssef M. Roman is focused on the effects of serum urate levels on human health, its relations to various diseases and the potential therapeutic benefits of modifying uric acid levels. The topic of this article is of interest to a broad audience including physicians, physiologists, pharmacologists, and evolutionary biologists. Primate during evolution progressively lost the uricase enzyme activity and the lack of this enzyme activity has been proposed to be advantageous as uric acid in serum plays various roles, including antioxidation, maintaining water in the body, inducing immune response, which could be beneficial to surviving from pathogen infection, and stimulating fructose metabolism and fat accumulation, which is believed to be important historically for hominoids to survive during climate changes and famines. By thoroughly reviewing literature, the author stated that abnormal levels, both high and low levels of uric acid may cause different human diseases thus modifying the level of uric acid, the end-product of purine metabolism is indeed a double-edged sword. High uric acid level has been well known correlating with gout disease and a growing population of hyperuricemia and gout has been a concern of public health. This is also linked to disorder of sugar and lipid metabolism resulting in cardiovascular diseases and obesity. Low level of uric acid is not as well-known for its negative effect on human health. However, it has been linked to high risk of neurodegenerative disease such as Parkinson’s disease and cognitive impairment in Alzheimer’s disease, although the mechanism has not been well elucidated.

As the author stated “serum urate levels can be influenced by sex, obesity, genetics, and overall nutrition. Therefore, the measurement of uric acid may not accurately reflect the inherent urate level of the individual to draw a robust association between serum urate levels and the condition of interest...”, the manuscript appears suited for the theme of “personal medicine”. However, the manuscript could be significantly improved prior to recommended publication in Journal of Personalized Medicine.

Author’s Response: The author would like to thank the reviewer for their time and thorough review of the article. The author strongly believes that this review article has greatly improved because of the reviewer’s comprehensive assessment.

Major issues:

  1. The review article should include figures or tables for clearer presentation.

For example, a figure/table to illustrate various hypotheses, benefits and underlying mechanisms regarding the progressive loss of uricase during evolution of primates would be a nice addition.

Results of the GWAS study that revealed major determinants of uric acid levels and modulators of gout risk could be summarized in a table listing the gene names and encoded enzymes/ proteins and underlying mechanisms. As the manuscript in its current form, the gene acronyms listed without definition/explanation are not informative. “SLC2A9, SLC16A9, SLC17A1, SLC22A11, SLC22A12, ABCG2, PDZK1, GCKR, LRRC16A1, HNFA4G, and INHBC”

The complex relationship between fructose metabolism and loss of uricase/uric acid levels and underlying mechanisms could be presented as a figure.

What human diseases/disorders and how they are linked to either high or low levels of uric acid can be better summarized in a table.

Author’s Response: Thank you for your comments and suggestions. The author agrees and provides three figures and a table addressing the reviewer’s comments and suggestions.

  1. The paragraphs regarding fructose metabolism is very confusing as it is currently written, with several observations/hypotheses mixed and messed up together.

How uric acid stimulates fructose metabolism and hence adipose tissue formation is not clearly stated (Lanaspa et.al PLoS One. 2012;7(10):e47948.). Indeed, it acts through the transcriptional upregulation of the fructokinase. And this should be used as part of the evidence for hominoids employing loss of uricase as a survival mechanism for fat accumulation during famine period.

Author’s response: Thank you for the comment and suggestions. The author agrees with the reviewer’s comments. I believe Figure 2 does help to clarify this point. Also, the author revised the paragraph to address the reviewer’s comment. Please see lines 167-176

“Additionally, increased uric acid levels, as a result of declining uricase activity, could have amplified the effect of fructose on energy and fat storage (Figure 2). While high serum urate levels may enhance the development of metabolic syndrome, uric acid could further potentiate the metabolism of fructose by increasing the expression of fructokinase, which in turn can increase the formation of fructose 1-phosphate and the formation of energy storages such as glycogen and triglycerides (Figure 2).[36]”

The author stated “Fructose induces insulin resistance partly due to glucogenesis mediated by fructokinase-dependent activation of the AMP deaminase pathway with the generation of uric acid”. This does explain why excessive fructose intake is linked to high uric acid level. However, this statement is placed in a wrong paragraph where the author wants to interpret the reason behind loss of uricase. Please note that we have already lost uricase enzyme activity millions of years. Nevertheless, excessive fructose intake has raised broad concern and it is generally believed to be a cause of hyperureicemia and gout. thus it is worth detailing the biochemical mechanism how even without uricase this would raise the uric acid level by increasing the rate of purine degradation through the activities of fructokinase and AMP deaminase.  

 Author’s response: Thanks for your comment. The author partly agrees with the reviewer’s comment. The purpose of this paragraph is two-fold. First, the author would like to present the biochemistry behind fructose metabolism which leads to high uric acid. Second, the author would like to present how fructose was important as a survival switch for hominoids to overcome famine and food scarcity. The author agrees with the reviewer’s comment that AMP deaminase is activated by the fructokinase which further increases uric acid levels. The paragraph was slightly edited to account for the reviewer’s comments. Please see lines 194-201.

“Moreover, fructose could induce insulin resistance, partly due to glucogenesis mediated by fructokinase-dependent activation of the AMP deaminase pathway with the generation of uric acid (Figure 2).[39] With glucose being the primary source of immediate energy needs, especially the brain, preserving an adequate supply of glucose is crucial for survival. Fructose reduces resting metabolism and stimulates fat and glycogen accumulations, as a means to reduce the metabolism of and preserve the levels of glucose, which could well explain the insulin resistance induced by fructose (Figure 3).[14] These metabolic processes could have enabled primates and humans to survive starvation during significant climate changes, including restricted water access, droughts, or devegetation.”

“Indeed, it is believed that with loss of the vitamin C encoding gene due to significant ingestion of fruits and vegetables, the primary source of vitamin C, uric acid became the primary antioxidant and free radical scavenger when consumption of fruits and vegetables dramatically decreased due to substantive climate changes.”-----This sentence is not comprehensible once again this is a mixture of different evolutionary events and hypotheses. Please rephrase it. Despite most vertebrates capable of making vitamin C, hominoids are not the only ones that can not make it. i.e, teleost fishes, guinea pigs, and some bats. Do all these vertebrates use uric acid as a radical scavenger? Did they also encounter climate changes and food shortage and lose their uricase?

 Author’s response: Thank you for your comment and interesting question. The reviewer is correct about the hypothetical nature of this relationship between diet and vitamin C biosynthesis. Therefore, the author made it clear to distinguish between the facts and hypothesis-driven comments. Regarding the reviewer’s question, there is no reason to think that the antioxidant activity of uric acid will be different between species. However, the author believes that the high concentration of uric acid in hominoids not only provides antioxidant activity that replaces vitamin C but also enables hominids to survive energy shortages that vitamin C couldn’t do.

It is hypothesized that the loss of the L-gulono-γ-lactone oxidase gene, which is involved in vitamin C biosynthesis, was driven by the significant ingestion of fruits and vegetables, a primary source of vitamin C. As a result, uric acid became the primary antioxidant and free radical scavenger when fruit and vegetable consumption dramatically decreased due to substantive climate changes. Even though most vertebrates can make vitamin C, hominoids are one of the few species that cannot make it. These observations suggest that retaining high uric acid levels may have enabled hominoids to garner survival benefits above and beyond the antioxidant activity of uric acid.

  1. Diet has been attributed several times to gout onset throughout the text, more important than genetics in Line 70 for Maori of New Zealand who has gout susceptible alleles, more important than local environments in line 246 for Phillipinos and Japanese (compared with native residents). Diet including fructose, fatty meats and salt have all been mentioned. All gout-related diet information scattered throughout the text and appeared very disorganized. Please present it in a more thoughtful way.  

 Author’s response: Thank you for the comment. I believe Figure 1 provides some summary of the contribution of diet and culture and the risk of gout and gout-related comorbidities.

  1. As the author wrote in the abstract “…provide new insights into the potential therapeutic benefits of uric acid and novel uricase-based therapy.” Surprisingly, there is no discussion of krystexxa, or any other uricase-based therapy at all in the text.

Author’s response: The author would like to thank the reviewer for the comment. A more focused

paragraph on currently available recombinant uricase drugs was added in the background section.

Please lines 35-44

For the purpose of increasing uric acid, the author spent quite a large volume on SLC2A9 without any definition/description of the gene and the coded transporter until at the very end defining it as GLUT9. I would recommend a clearer definition and description of the transporter and its functional relationship with URAT1 and how both are targets of the anti-gout drug, benzbromarone. So to increase uric acid level, one would likely need an activator of GLUT-9? is such a compound already available?  The authors listed many determinants of uric acid levels. SLC2A9, SLC16A9, SLC17A1, SLC22A11, SLC22A12, ABCG2, PDZK1, GCKR, LRRC16A1, HNFA4G, and INHBC…. Could they serve as targets to increase or decrease uric acid levels? 

 Author’s response: Thank you for the comment. The author agrees with the reviewer’s comment. A detailed table was provided to describe the gene, protein, and function. Please see Table 1. As far as raising uric acid levels, most of the literature has focused on xanthine oxidase inhibitors or using uric acid precursors. To the author’s knowledge, the role of targeting urate transporter as a target to increase urate levels has not been explored.

Other than urine through kidney, gut provides the excretion pathway for the remaining 1/3 of uric acid. A recent paper described a gene cluster in human gut microbiome that compensates for loss of uricase (Liu et al., 2023, Cell 186, 3400-3413). Another research identified the long-sought anaerobic purine degradation pathway that bypasses the formation of uric acid (Tong et al., 2023, Cell Chemical Biology 30, 1-11). Given the importance of diet-derived purine, probiotics engineered with this pathway is potentially an efficient modifier of the uric acid level of the serum. Both papers should be cited.

Author’s response: The author would like to thank reviewers for pointing out these important papers. The papers were cited and a brief description of the role of the microbiome was added to the background section. Please see lines 27-37

“In humans, two-thirds of uric acid is primarily excreted through the kidney and one-third through the gastrointestinal tract.[3] While humans do not produce uricase per se, a growing body of literature has suggested that the gut microbiome, a major source of uricase, may have a role in for compensating the loss of the uricase gene.[4-7] Increased urate production or urate underexcretion can lead to increased serum urate levels beyond the solubility threshold, resulting in the formation of monosodium urate (MSU) crystals in and around the joints. Mobilization of MSU crystals could trigger an inflammatory response known as gout flares.”

Minor changes requested:

  1. “Markedly high uric acid levels in humans have been caused by the loss of the uricase-encoding gene during the Neogene period” “Specifically, the lack of uricase activity in humans has been greatly attributed to the nonsense mutation in codon 33 in exon 2, a nonsense mutation in codon 187, and a splice mutation in exon 3.”

These two sentences contradict with each other.. “loss of the uricase-encoding gene” this expression is not accurate… I would suggest loss of the enzyme/enzyme activity.

Author’s response: Thank you for the comment. The suggested language was incorporated into the paragraph. Please see line 26.

  1. The definition of an abbreviation should be made in its first appearance and then the abbreviation can be used in later paragraphs. for example after definition, “monosodium urate” does not need to be spelled out and can be replaced by MSU

Author’s response: Thank you for the comment. The abbreviations were used when appropriate.

  1. “vitamin C encoding gene” ---this is a wrong expression.. genes encode proteins i.e L-gulono-γ-lactone oxidase that is involved in vitamin C biosynthesis.

Author’s response: Thank you for the comment and correction. The author made the necessary revisions. Please see lines 77-81

  1. “cerebrovascular events” including “acute ischemic stroke events” may lead to the development of a neurodegenerative-like condition. But stroke is not classified as neurodegenerative disease.

Author’s response: Thank you for the comment. Though stroke is not classified as a neurodegenerative disease, it significantly increases the risk of developing neurodegenerative diseases. However, to facilitate the flow of the manuscript, the author moved the section on stroke to hypertension. Please lines 284-289.

Reviewer 2 Report

Roman YM presents a review article that delineate the importance of uric acid on human health. The author elucidates the uricase enzyme's role and the physiological significance of high and low levels of uric acid on human health. In addition, author has discussed how the level of uric acid in our serum could be beneficial and harmful in the context of neurodegenerative diseases. This review article has covered most of the aspects associated with uric acid. However, I have a few concerns which potentially would improve the manuscript.

Comments-

1.       A pictorial representation of the effects of uric acid on human health is missing. Cartoon graphs help readers to understand the summary of the article. The author should add figures to explain the importance of uric acid in human health for the respective sections of the article.

2.       Author has discussed the correlation of hyperuricemia with SLC2A9, which encodes GLUT9 transporter, but did not discuss the physiological connection of GLUT9 transporter with uric acid anywhere in the article.

3.       References for citations are added after the period throughout the manuscript.

Author Response

Reviewer 2:

Roman YM presents a review article that delineate the importance of uric acid on human health. The author elucidates the uricase enzyme's role and the physiological significance of high and low levels of uric acid on human health. In addition, author has discussed how the level of uric acid in our serum could be beneficial and harmful in the context of neurodegenerative diseases. This review article has covered most of the aspects associated with uric acid. However, I have a few concerns which potentially would improve the manuscript.

Author’s response: The author would like to thank the reviewer for reviewing the article and providing constructive feedback.

Comments-

  1. A pictorial representation of the effects of uric acid on human health is missing. Cartoon graphs help readers to understand the summary of the article. The author should add figures to explain the importance of uric acid in human health for the respective sections of the article.

Author’s response: Thank you for the comment and suggestion. The same suggestion was made by the other reviewer. The author provided three figures, a graphical abstract, and one table to address your comments.

  1. Author has discussed the correlation of hyperuricemia with SLC2A9, which encodes GLUT9 transporter, but did not discuss the physiological connection of GLUT9 transporter with uric acid anywhere in the article.

 Author’s response: Thank you for the comment. You are correct. The author provides a table describing the role of various genes involved in uric acid, including SLC2A9. Please see table 1.

  1. References for citations are added after the period throughout the manuscript.

Author's response: Thank you for the comment. I will be happy to address this concern at the production stage of the manuscript.

Round 2

Reviewer 2 Report

Author has addressed my comments.